# Development and Experimental Study of Mobile Fire Smoke Decontamination System

Hongyong Yuan [1], Yang Zhou [1], Fan Zhou [1,2,*], Lida Huang [1] and Tao Chen [3]

[1]   Hefei KDLian Safety Technology Co., Ltd., #5999 Xiyou Road, Hefei 230088, China
[2]   Hefei Institute for Public Safety Research, Tsinghua University, #5999 Xiyou Road, Hefei 230601, China
[3]   Department of Engineering Physics, Tsinghua University, 30 Shuangqing Road, Beijing 100190, China
*   Correspondence: zhoufan2@mail.ustc.edu.cn; Tel.: +86-551-63601641; Fax: +86-551-63606459

**Abstract:** Fire smoke decontamination equipment, such as fire-fighting robots and smoke exhaust robots, is mainly used in long and narrow spaces such as underground garages. In several recent decades, the study of fire smoke spread in narrow spaces and fire smoke decontamination equipment stimulated the interests of many researchers. However, present equipment cannot eliminate insoluble toxic gases such as CO and may decrease the height of the smoke layer, causing great difficulty to rescue. In this study, a novel mobile fire smoke decontamination process and system are proposed. The experimental study and theoretical prediction of the system are conducted. The results show that the developed equipment is able to eliminate fire smoke particles and CO, cool the space, and improve the visibility of the fire site. The developed equipment can reduce the space temperature to below 60 °C, reduce the CO concentration to below 145 ppm, and enhance the visibility to more than 50 m in the rectangular tunnel after operating for 30 min under 4 MW fire condition.

**Keywords:** fire-fighting robot; smoke exhaust robot; fire smoke decontamination; long and narrow space





## 1. Introduction

With the development of urbanization, traffic tunnels, underground buildings, and other key urban infrastructures have become more and more complex. Such places belong to long and narrow urban spaces [1–3]. In these places, fire risk is exceptionally high and can easily cause huge casualties and property losses [4,5]. The difficulty of fire smoke control in these narrow and long urban spaces is mainly reflected in the high smoke concentration, high space temperature, and great difficulty in evacuation and rescue.

There has been much research concerned with fire smoke control difficulty in long and narrow spaces in the past several decades. Common smoke control methods include longitudinal ventilation [6,7], semitransverse ventilation [8], natural smoke exhaust in shafts [9], lateral smoke exhaust [10], centralized smoke exhaust [11], and a combination of multiple smoke exhausts [12]. The longitudinal ventilation system is widely used in tunnel ventilation systems due to its simple structure and low cost. However, past research on fire smoke control in narrow and long spaces is based on the fixed firefighting system [13,14], and there is a lack of movable smoke decontamination methods for smoke control in the fire site. Compared with the fixed firefighting system, the mobile firefighting system can reach the fire site and start working, which can quickly control the fire in the early stage. Therefore, a mobile smoke disposal robot (firefighting robot) has attracted researchers' attention.

Firefighting robots are mainly used to spray a large amount of water or water mist on the fire smoke to achieve the effect of cooling and smoke purification. Dinh [15] proposed a flying robot using waterpower and a novel weight-shifting mechanism. Yu [16] discusses the control system of a fire rescue robot for a high-rise building design. A study [17]

proposed an indoor firefighting robot that has the capability to climb stairs and negotiate several types of floor materials inside buildings. Lindawani [18] designed and made a robot control system hardware for firefighting-legged robots, aimed to make the robot have a better obstacle-surmounting ability in the fire site. Furthermore, there are many studies that concentrate on the fire and smoke suppression mechanism for firefighting robots.

In long and narrow spaces, especially underground spaces, fixed smoke exhaust equipment usually cannot effectively decontaminate a large amount of fire smoke. The application of firefighting robots will greatly improve the ability to extinguish the fire and implement an effective fire rescue, especially in very dangerous situations or when the fire site is inaccessible to firefighters. However, present tunnel firefighting robots still face many technical difficulties [19], especially since they cannot eliminate insoluble toxic gases such as CO and may destroy the smoke layer structure and decrease its height, thus causing great difficulty to rescue.

In this study, a novel mobile fire smoke decontamination process and system are proposed which can actively inhale smoke and conduct decontamination inside. The proposed system is able to eliminate fire smoke particles and CO, cool the space, and improve visibility in long and narrow spaces without destroying the smoke layer structure. The experimental study and theoretical prediction of the system are conducted.

## 2. Fire Smoke Decontamination Process and System

### 2.1. Fire Smoke Decontamination Process

The function of the mobile fire smoke decontamination system (FSDS) in this study is set to decontaminate fire smoke in confined spaces such as highway tunnels and underground garages. The FSDS is mounted on a mobile platform that can transport the FSDS to the ignition point or personnel evacuation exit automatically. The FSDS then starts fire smoke decontamination to eliminate toxic substances in space, reduce space temperature, and improve visibility. In this study, the FSDS is designed to eliminate fire smoke particles and CO/soluble toxic substances and cool the space. The design parameters of the FSDS working fire environment are listed in Table 1.

**Table 1.** The design parameters of the FSDS working fire environment.

| Design Parameters | Value |
|---|---|
| Fire location | Highway tunnels/underground garages |
| Fire type | Automobile fire |
| Fire power | 0~6 MW |
| Smoke production | 0~4000 $m^3$/h |
| Space temperature | 60~400 °C |
| CO concentration | 500~2000 ppm |

Figure 1 shows the fire smoke decontamination process of the FSDS. The fire smoke is firstly induced into the inlet of the FSDS due to the negative pressure environment produced by the centrifugal fan. Next, the fire smoke is transmitted to the CO filter through a titanium alloy pipe with high temperature resistance which has no deformation under a 1000 °C fire smoke atmosphere. After CO elimination, the fire smoke will experience particulate filtration. In this process, most of the particles larger than 20 microns will be filtered. During the process of smoke inducement, CO elimination, and particulate filtration, the core module maintains a negative pressure state, which can prevent the fire smoke from escaping. The centrifugal fan pressurizes the primary exhaust to a positive pressure state. Then, the mixed cooling heat exchanger will cool down the fire smoke using the latent heat cooling method and exhaust the clean smoke to space.

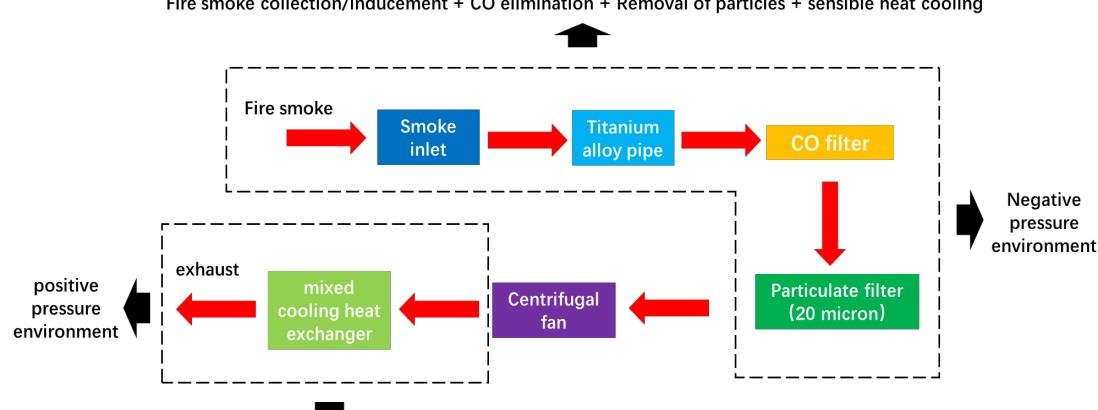

**Figure 1.** Fire smoke decontamination process.

### 2.2. Fire Smoke Decontamination System

Figure 2 shows the design of the FSDS. The smoke inlet is designed to be a flaring shape. The electric rotator is added at the end of the titanium alloy pipe in order to control the smoke inlet position. The CO filter adopts a metal catalytic oxidation CO elimination design, which can convert CO into carbon dioxide with 90% efficiency. The particulate filter uses an alloy (iron chromium aluminum) fiber wire metal filter element to filter carbon black particles and oil droplets in smoke. In total, 95% of particles larger than 20 μm will be eliminated. The accumulated ash on the filter element will be cleared by the back-blowing purification system according to the measured $\Delta P$ (inlet–outlet) of the particulate filter. The mixed cooling heat exchanger adopts water spray (particle size: 250 μm) to cool down the smoke flow from the centrifugal fan and filter the soluble toxic substances simultaneously. During the smoke cooling, the water spray will be partially (70~80%) evaporated into water vapor, which can absorb a lot of latent heat from the smoke. The design parameters of the FSDS are listed in Table 2. Figure 3 shows the design drawing and physical drawing of the FSDS with mobile devices.

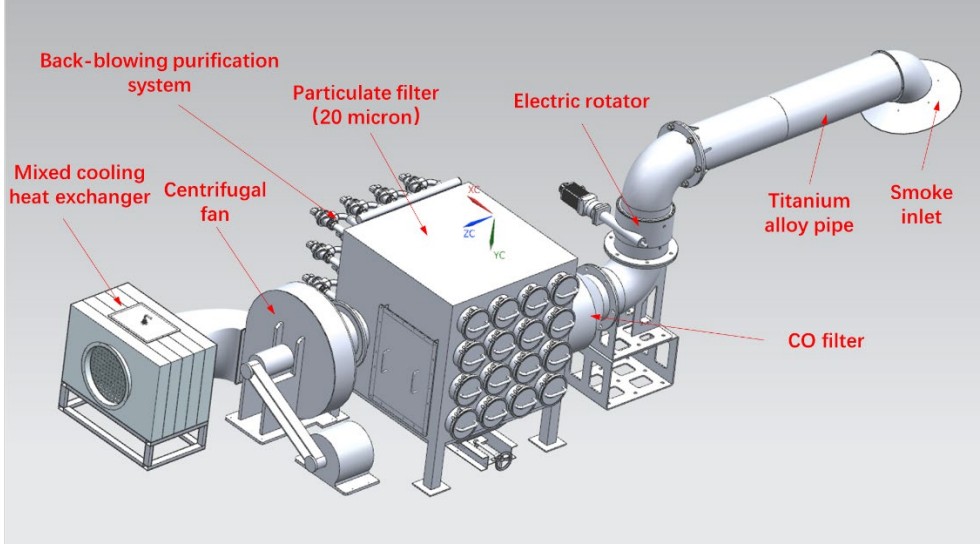

**Figure 2.** Design of the fire smoke decontamination system.

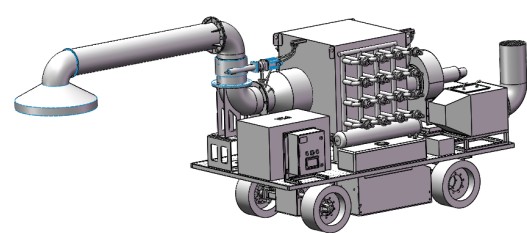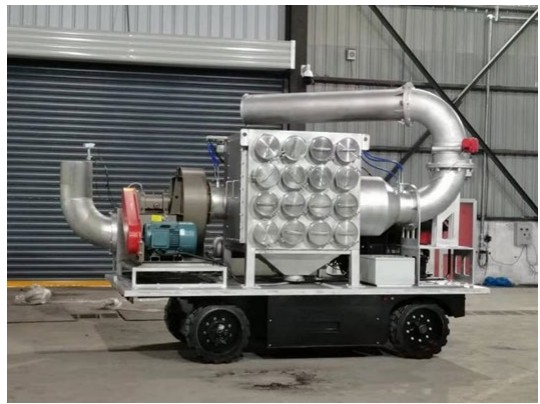

**Figure 3.** Design drawing and physical drawing of the FSDS with mobile device.

**Table 2.** The design parameters of the FSDS with mobile device.

| Design Parameters | Value |
|---|---|
| Fire smoke decontamination capacity | 9000 m$^3$/h (Cold operation)/5000 m$^3$/h (normal operation) |
| CO filtration efficiency | 90% |
| Particulate filtration size/efficiency | 20 um/95% |
| Smoke cooling power | 500 KW |
| Water spray flux/pressure | 300 Lh$^{-1}$/0.5 Mpa |
| Overall power | 13 KW |
| Mobile speed | >5.4 km/h |
| Climbing angle | <15° |
| Turning radius | <3 m |
| Surmountable obstacle height | 120 mm |

## 3. Description of the Experiment

### 3.1. Experimental Set Up

In order to validate the overall effect of fire smoke decontamination of the FSDS, a validation experiment set up was built. Figure 4 shows the design drawing and physical drawing of the experiment setup. The oil pool fire and automobile tire combustion are adopted to simulate the fire occurrence process (heat release rate: 1~1.5 MW). The experimental site is selected to be in the large space fire test platform in Hefei, China. After the oil pool device is ignited, the FSDS is started and kept 10 m away from the fire source until the smoke is stably generated. During the waiting period, the cold operation (without fire smoke) performance of the FSDS is tested. Then, the FSDS moves to the top of the fire source to start smoke decontamination and normal operation data collection. The decontamination test period lasts 10–20 min.

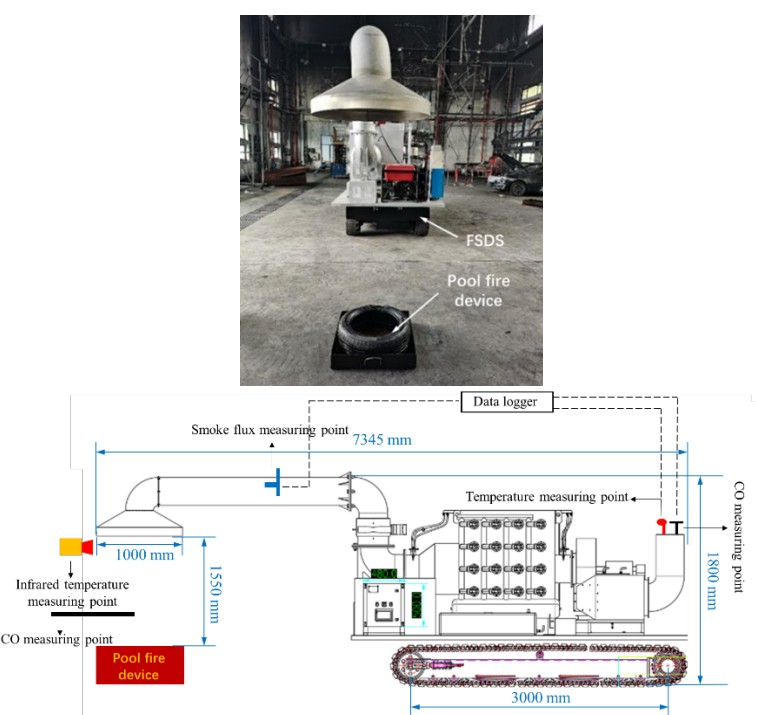

**Figure 4.** Designing drawing and physical drawing of the experiment setup.

### 3.2. Sensors Arrangement and Data Collection

Figure 4 also shows the data measuring points of the FSDS. The inlet temperature is measured by the infrared temperature measuring device. The exhaust temperature is measured by the Thermocouple (K type). The inlet/exhaust CO concentration is measured by the handheld CO detector and Duct type CO sensor, respectively. The smoke flux is transferred from the smoke flow velocity, which is measured by the pitot tube inserted into the titanium alloy pipe. Table 3 shows the list of the main experimental measuring apparatus and accuracies.

**Table 3.** List of the main experimental measuring apparatus.

| Apparatus | Measuring Parameter | Accuracy |
|---|---|---|
| Thermocouple (K type) | Temperature (FSDS exhaust) | $\pm 0.5\,°C$ |
| Duct type CO sensor | CO concentration (FSDS exhaust) | FS $\pm$ 1% |
| Pitot tube (AFP-8A) | Smoke flow velocity (FSDS titanium alloy pipe) | FS 0.02% |
| Infrared temperature measuring device (FLUKE-MT4) | Temperature (FSDS inlet) | FS $\pm$ 2% |
| Handheld CO detector (FZ-BX) | CO concentration (FSDS inlet) | FS $\pm$ 3% |
| Data logger (LR8431-30) | / | / |

### 3.3. Evaluation of the Operating Performance of the FSDS

In the experiment, four performance parameters are adopted to evaluate the overall operating performance of the FSDS, which are the overall smoke decontamination volume flux $F_n$, the cold operation decontamination flux $F_c$, the CO filtration efficiency $\eta_{CO}$, and the smoke cooling heat exchange power $P_c$. The mathematical expressions for the four parameters are as follows:

$$F_n = V_n S_{tap} \times k_t$$

$$F_c = V_c S_{tap} \times k_t \eta_{CO} = \frac{C_{co-inlet} - C_{co-outlet}}{C_{co-inlet}}$$

$$P_c = \rho_s F_n (T_{inlet} - T_{outlet})$$

where $S_{tap}$ is the cross-section area of the titanium alloy pipe, $k_t$ is the conversion factor between standard volume (under 0 °C, 0.1 Mpa) and measuring volume [20], $C_{co-inlet}$ and $C_{co-outlet}$ are CO concentrations tested in the inlet and outlet, respectively, and $\rho_s$ represents the density of the fire smoke in the titanium alloy pipe.

## 4. Tunnel Fire Simulation

In this work, the fire simulation software FDS is used for the calculation of the FSDS operation process and semitransverse ventilation process in the rectangular tunnel. FDS is developed by the NIST Building Fire Laboratory of the United States. The software adopts the field simulation method to discretize and iterate the control equations related to smoke spread, fire combustion, heat, and mass transfer in the calculation area. Nonuniform mesh, gradually sparse from the wall to the interior area, is used for mesh division in the theoretical model. The average grid size of the cross-section area of the tunnel is 20 mm, and the average grid size of the longitudinal direction of the tunnel is 100 mm. The grid sensitivity test is conducted through choosing a typical simulation condition. The results are show in Table 4. It can be seen that the selected mesh in the model is able to draw the simulation results with high accuracy.

**Table 4.** The results of grid sensitivity test.

| Size of the Grid | Simulation Result of the Average Smoke Layer Height |
|---|---|
| 20 mm (cross-section); 100 mm (longitudinal) | 4.51 m |
| 10 mm (cross-section); 500 mm (longitudinal) | 4.35 m |
| 15 mm (cross-section); 150 mm (longitudinal) | 4.54 m |

### 4.1. Simulation for the FSDS Operation Process

To evaluate the FSDS operation performance in real confined spaces (fire environment), a simulation model for the FSDS operation process in the rectangular tunnel is present in this part. The simulation model includes fire smoke spread, solution of temperature/CO concentration distribution, visibility calculation, and FSDS smoke decontamination process. The simulation model is based on the following major assumption:

The operating performance of the FSDS, including $F_n$, $F_c$, $\eta_{CO}$, and $P_c$, remains constant during the smoke decontamination.

Figure 5 shows the sketch of the simulation model for the FSDS operation process in the rectangular tunnel. The size of the rectangular tunnel is as follows: length 200 m, width 10 m, and height 7 m (14,000 m$^3$). The boundary conditions at both ends of the tunnel are set as "Open" type. The fire source (4 MW) is located in the center of the tunnel (fire power of single car). The smoke outlet of the FSDS is located directly above the fire source, with a height difference of 1 m from the fire source. The FSDS operation parameters are listed in Table 5.

**Table 5.** List of the measured fire environment parameters.

| Parameters | Value |
|---|---|
| Average environment temperature | 11.3 °C |
| Average FSDS inlet temperature | 230.0 °C |
| Average FSDS inlet CO | 1610 ppm |

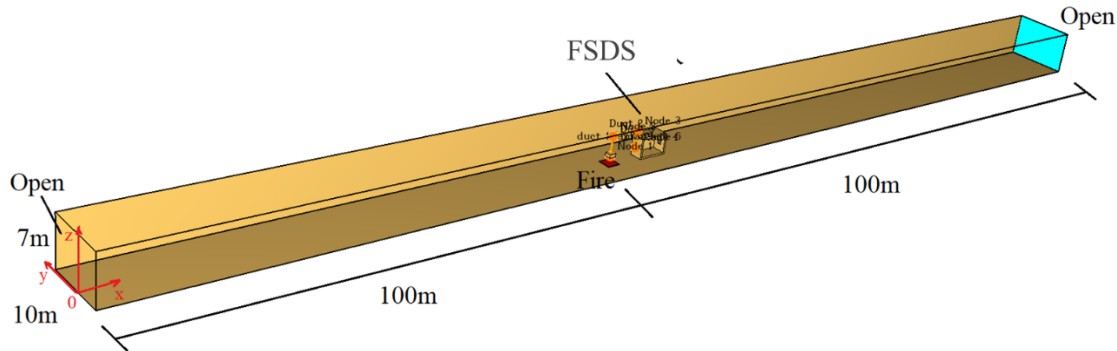

**Figure 5.** The sketch of the simulation model for the FSDS operation process in rectangular tunnel.

Figure 6 shows the arrangement of temperature, CO concentration, and visibility measuring points in the rectangular tunnel. The height of each measuring point is 1.8 m. Three rows of measuring points are arranged along the width of the tunnel at y = 0 m, y = −2 m, and y = −4 m.

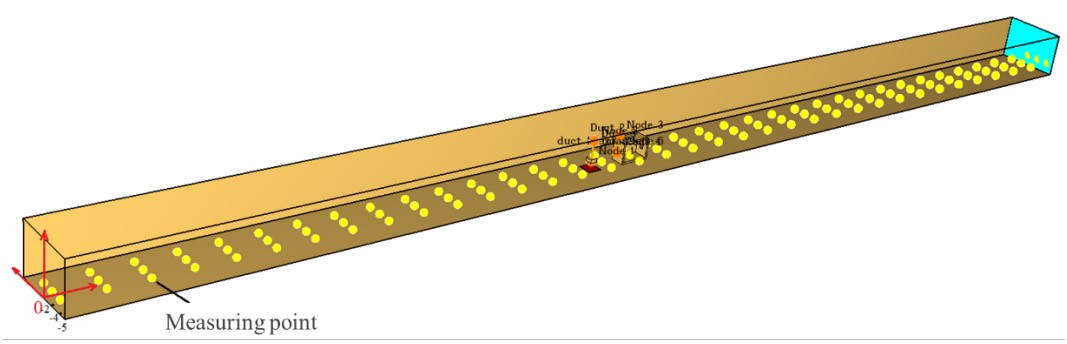

**Figure 6.** The arrangement of measuring points in the rectangular tunnel.

### 4.2. Simulation for the Semitransverse Ventilation Operation Process

To conduct the comparison study between the FSDS operation performance and semitransverse ventilation operation performance, a simulation model for the semitransverse ventilation operation process in the same rectangular tunnel is present in this part. Figure 7 shows the sketch of the simulation model for the semitransverse ventilation operation process in the rectangular tunnel. The fire source is located in the center of the tunnel. There are six mechanical smoke vents on the top of the tunnel, and the size of the smoke vents is 2 m × 2 m, the spacing is 30 m, and the total smoke exhaust volume is 190,000 $m^3$/h.

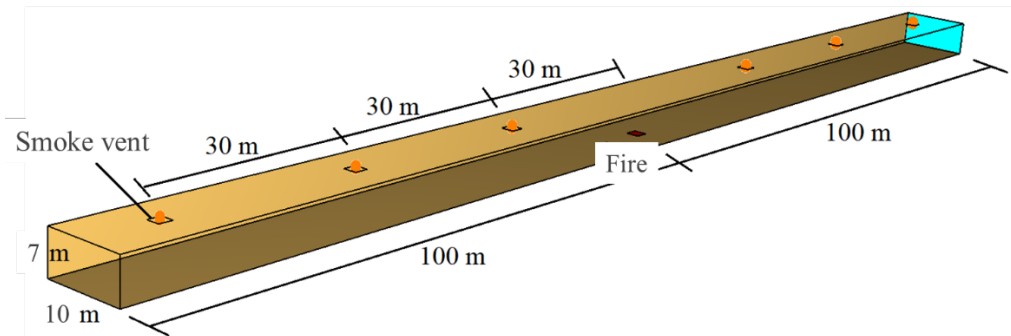

**Figure 7.** The sketch of the simulation model for the semitransverse ventilation operation process in rectangular tunnel.

## 5. Results and Discussions

In this section, the results of the FSDS fire smoke decontamination experiment are presented and analyzed. Performance parameters of the FSDS are calculated according to the experimental results. Based on the tested FSDS performance parameters, the simulation for the FSDS operation process in the rectangular tunnel is conducted. Longitudinal airflow in the tunnel, the height of the smoke layer, temperature/CO/visibility distribution at 1.8 m height, and inlet/exhaust temperature of the FSDS are calculated. the comparison study between the FSDS operation performance and semitransverse ventilation operation performance is conducted.

### 5.1. Experimental Results

The working state of the FSDS during the test is shown in Figure 8. It can be seen that most of the smoke generated by the fire source is inhaled by the smoke inlet of FSDS. The black degree at the smoke inlet is much higher than the outlet due to the fact that the exhaust includes decontaminated smoke water vapor and water particles.

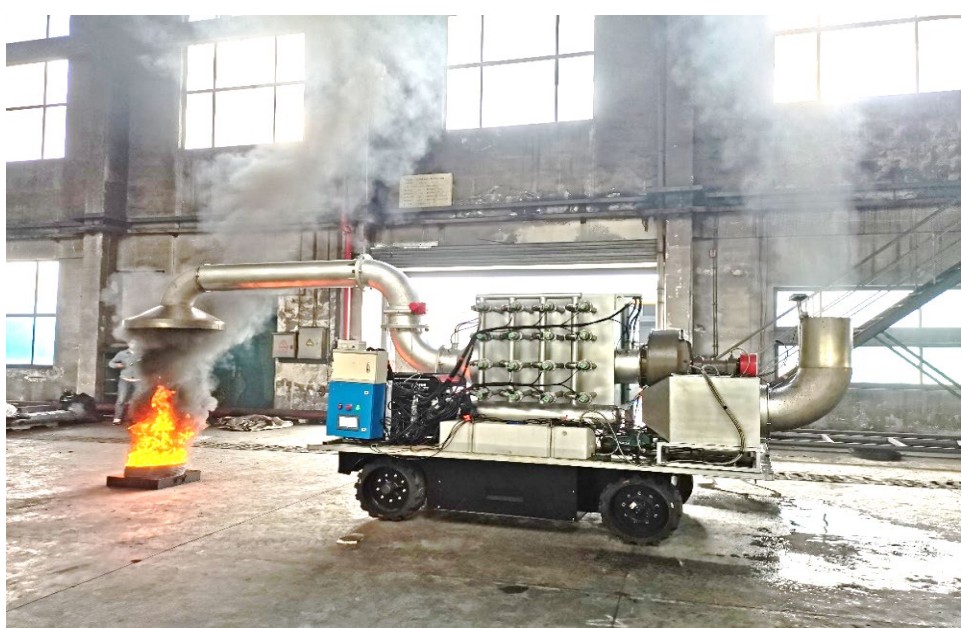

**Figure 8.** Working state of the FSDS during the test.

Table 5 lists the measured fire environment parameters. Figure 9 shows the exhaust volume flux variation of the FSDS. The exhaust volume flux keeps at around 8500 Nm$^3$/h before 80 s. This is because the FSDS keeps at a cold operation state before 80 s. The volume flux rises to around 8000 Nm$^3$/h at 300 s (start fire smoke decontamination). Then, the value continuously decreases to around 3000 Nm$^3$/h. This is due to the fact that the higher temperature inlet smoke has a higher specific volume, which leads to the lower operating performance of the centrifugal fan. The calculated $F_c$ and $F_n$ are listed in Table 6.

**Table 6.** List of the calculated FSDS operating performance parameters.

| Parameters | Value |
|:---:|:---:|
| $F_c$ | 8629.01 Nm$^3$/h |
| $F_n$ | 5472.37 Nm$^3$/h |
| $P_c$ | 417.7 KW |
| $\eta_{CO}$ | 78% |

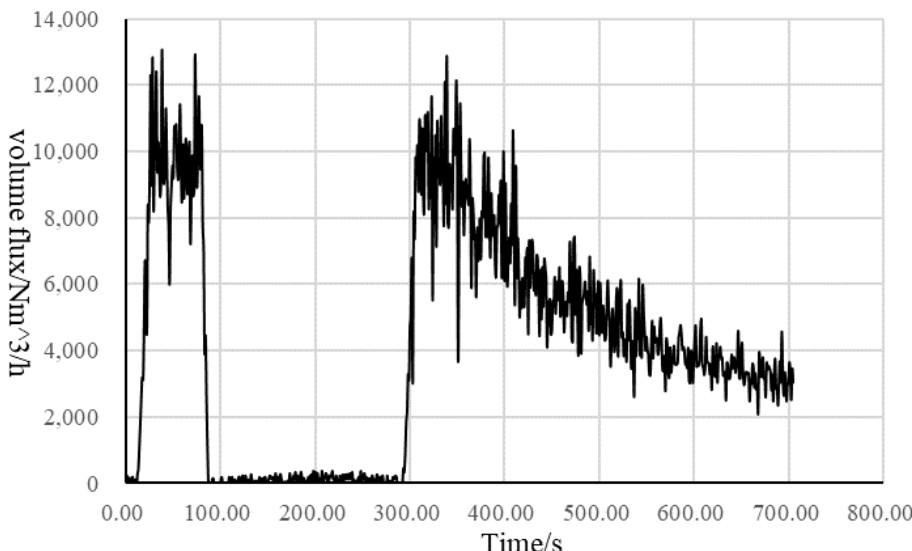

**Figure 9.** The exhaust volume flux variation of the FSDS.

Figure 10 shows the exhaust temperature variation of the FSDS. The exhaust temperature keeps at 12 °C before 300 s. This is because the FSDS keeps at a cold operation state before 80 s. After that, the exhaust temperature rises to 20 °C in the next 400 s. In terms of the measured average FSDS inlet temperature, the calculated $P_c$ is listed in Table 5. Figure 11 shows the exhaust CO concentration variation of the FSDS. It can be seen that the value begins to rise after the 300 s and decreases at the 600 s. The reason is that the CO filtration efficiency depends largely on the inlet temperature. The higher inlet temperature leads to a higher CO filtering efficiency within a certain range.

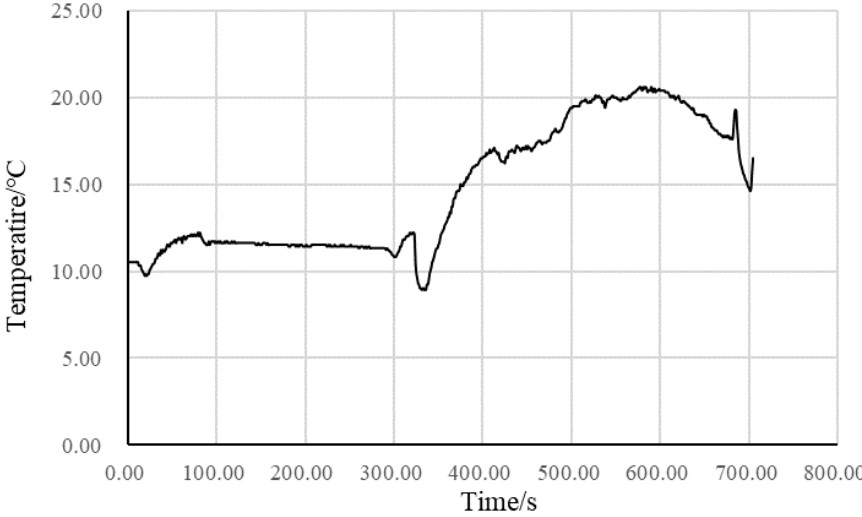

**Figure 10.** The exhaust temperature variation of the FSDS.

### 5.2. Performance Prediction of the FSDS in Rectangular Tunnel

In this section, longitudinal airflow in the tunnel, the height of the smoke layer, temperature/CO/visibility distribution at 1.8 m height, and inlet/exhaust temperature of FSDS are calculated to predict the contribution of FSDS to the fire rescue in the rectangular tunnel.

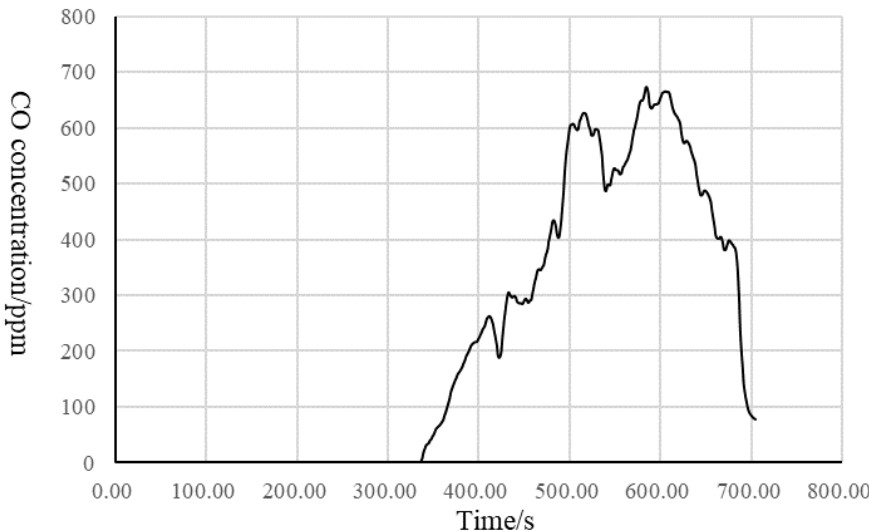

**Figure 11.** The exhaust CO concentration variation of the FSDS.

### 5.2.1. Longitudinal Airflow in Tunnel

The FSDS exhaust forms the induced longitudinal airflow along the length of the tunnel, as shown in Figure 12. Figure 13 shows the volume flow on the tunnel cross-section. The volume flow of the inducted airflow on the tunnel cross-section is about 20,000 m³/h. It can be seen from Figure 13 that the volume flow in the tunnel increases rapidly and tends to be stable after about 155 s.

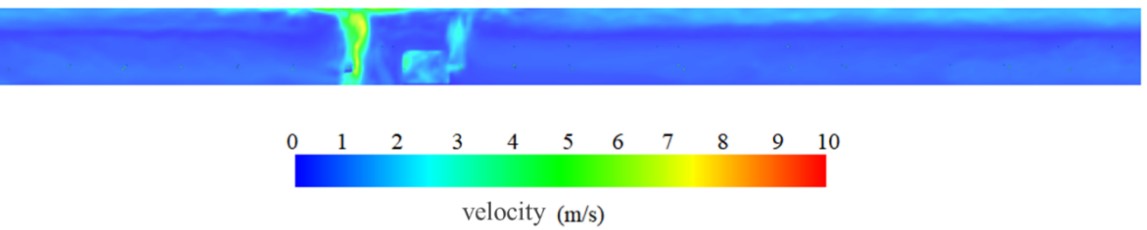

**Figure 12.** Longitudinal airflow velocity distribution in tunnel.

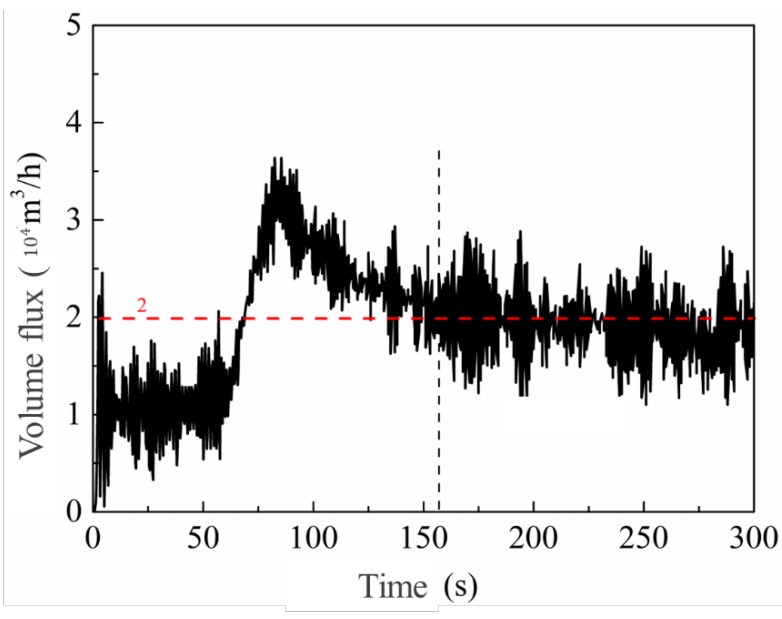

**Figure 13.** Volume flow on tunnel cross section.

### 5.2.2. Height of Smoke Layer

Figure 14 shows the height of the smoke layer in the tunnel. The height of the smoke layer near the fire source is about 5.2 m. The value in other areas is reduced due to the disturbance of the longitudinal airflow. It can also be seen from Figure 14 that the spray water volume of the FSDS has little influence on the smoke layer height. This indicates that the FSDS spray water will not destroy the structure of the smoke layer in the tunnel.

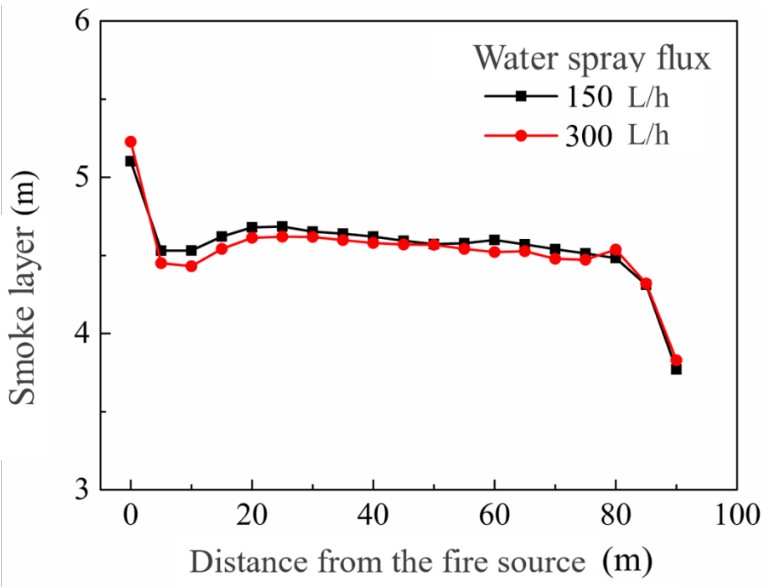

**Figure 14.** Height of smoke layer in tunnel.

### 5.2.3. Temperature/CO/Visibility Distribution at 1.8 m Height

Figure 15 shows the temperature distribution at 1.8 m height (average of 200–300 s in the stable section). It can be seen from Figure 15 that the temperature in the rest of the tunnel area is lower than 60 °C, except that the temperature near the fire source is higher. Figure 16 shows CO distribution at 1.8 m height. The CO concentration near the fire source is about 145 ppm. The value at 15 m away from the fire source is close to 0 ppm. Figure 17 shows the visibility distribution at 1.8 m height. It can be seen from the figure that the visibility at the fire source is the lowest, and the visibility in other areas is greater than 50 m.

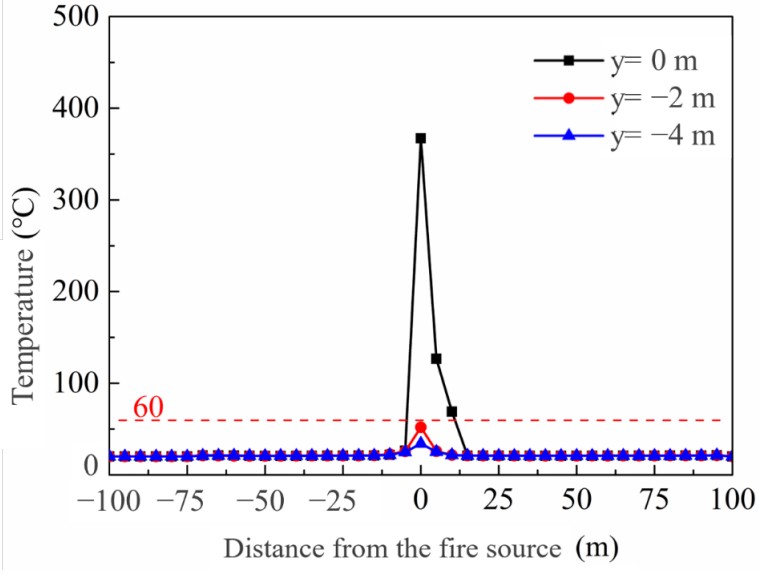

**Figure 15.** Temperature distribution at 1.8 m height.

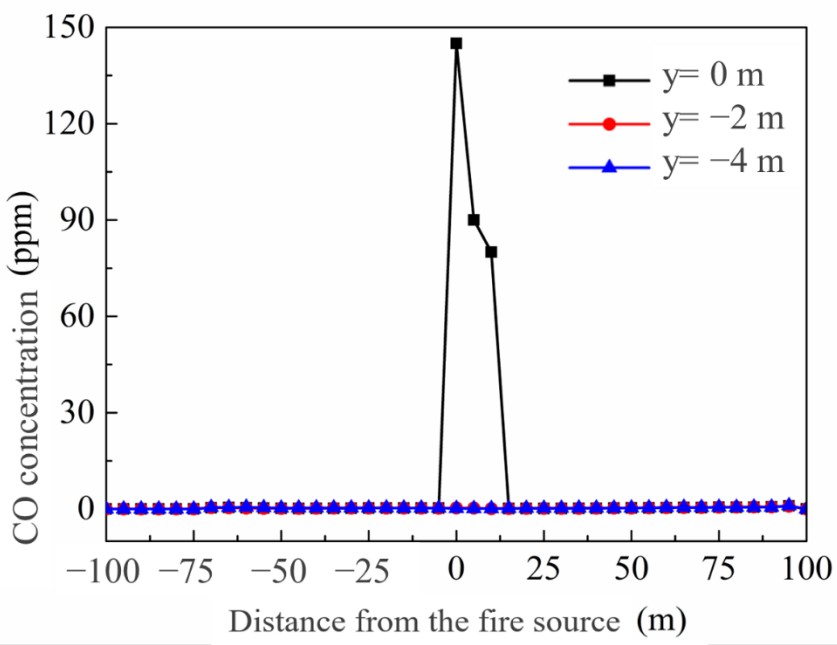

**Figure 16.** CO distribution at 1.8 m height.

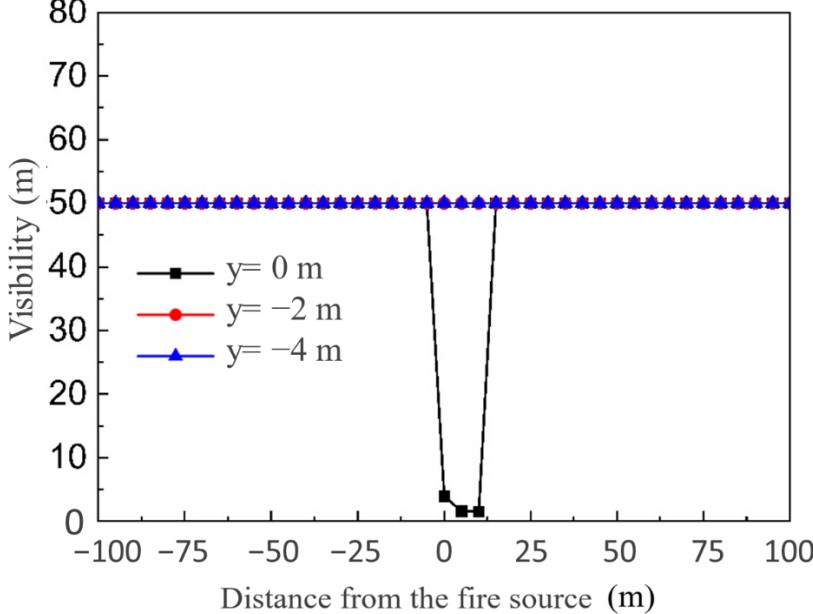

**Figure 17.** Visibility distribution at 1.8 m height.

5.2.4. Inlet/Exhaust Temperature of the FSDS

Figure 18 shows the inlet/exhaust temperature of the FSDS under different water spray fluxes. It can be seen that the temperature difference between the inlet and exhaust of the FSDS changes slightly with the change in the water spray flux. This indicates that 150 L/h is an ideal setting for the FSDS water spray flux.

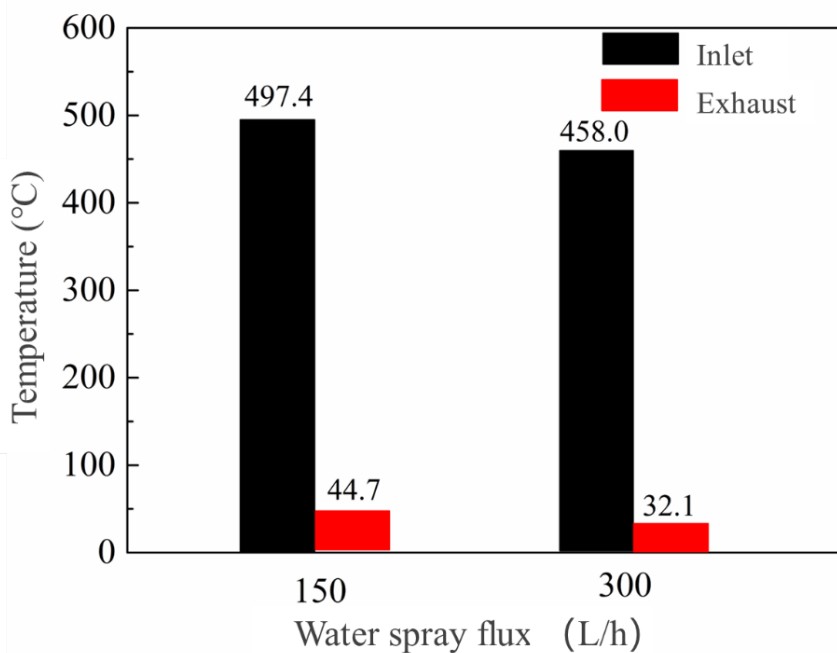

**Figure 18.** Inlet/exhaust temperature of FSDS.

*5.3. Semitransverse Ventilation Operation Performance*

5.3.1. Temperature Distribution at 1.8 m Height

Figure 19 shows the temperature distribution at 1.8 m height. It can be seen from the figure that the temperature at 1.8 m height decreases under the action of mechanical smoke exhaust. In contrast with the temperature distribution shown in Figure 15, the FSDS has the similar space cooling ability with semitransverse ventilation with the exhaust volume of 190,000 m$^3$/h. Both systems can maintain the space temperature below 60 °C in most of the tunnel, except for the place near the longitudinal smoke vents. The space cooling performance of semitransverse ventilation is higher than it is for the FSDS near the smoke vents.

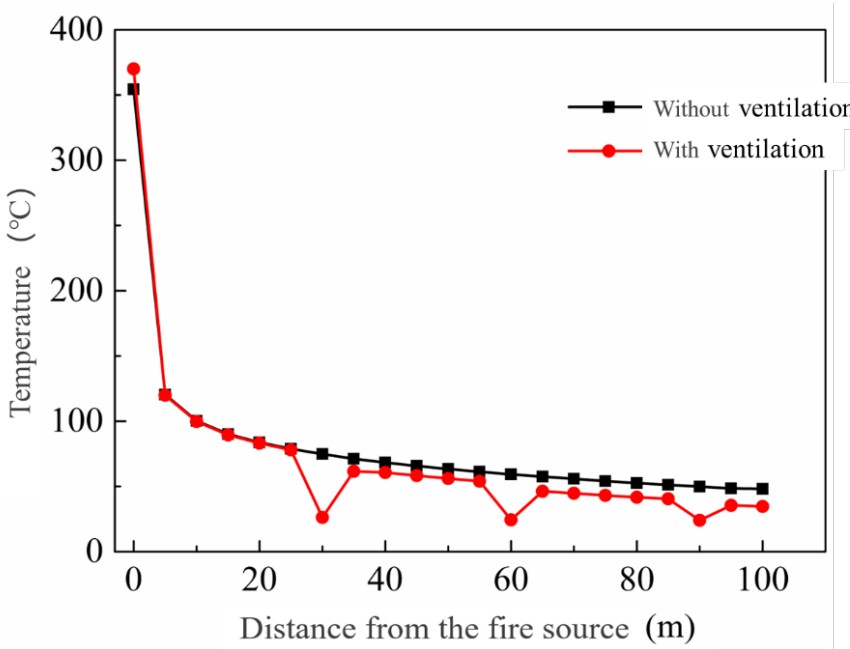

**Figure 19.** Temperature distribution at 1.8 m height (semitransverse ventilation).

5.3.2. Height of Smoke Layer

Figure 20 shows the height of the smoke layer in the tunnel. Compared with the working condition without smoke exhaust, the smoke layer height under the mechanical smoke exhaust condition is larger. This is due to the fact that smoke ventilation reduces the thickness of the smoke layer and increases the height of the smoke layer. Compared with Figure 14, it can be calculated that the average smoke layer height of the FSDS is 4.53 m, while the value for semitransverse ventilation system is 4.31 m. In consideration of the larger exhaust volume of the semitransverse ventilation system, it can be drawn that the FSDS has less damage to the smoke layer structure than the semitransverse ventilation system.

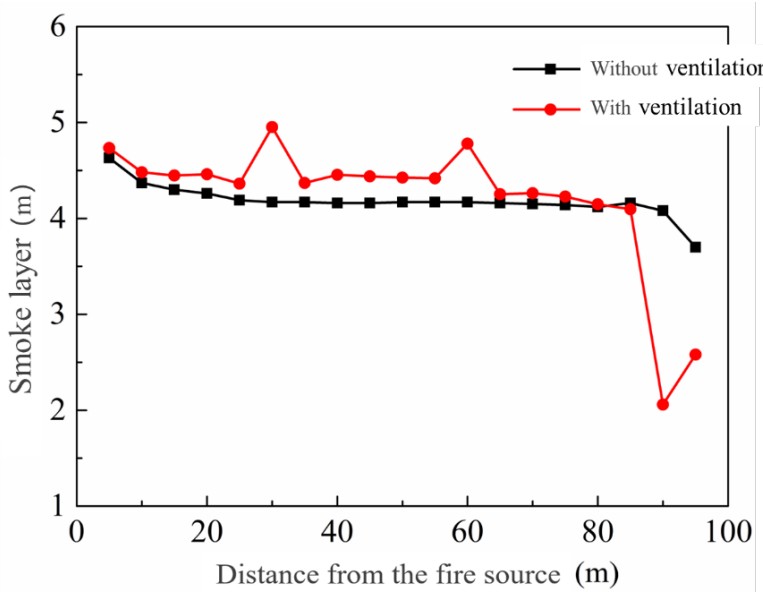

**Figure 20.** Height of smoke layer in tunnel (semitransverse ventilation).

## 6. Conclusions

In this paper, the development of a novel mobile fire smoke decontamination process and system are introduced. The experimental study and theoretical prediction of the system are conducted. The goal is to evaluate the FSDS working performance and its contribution to the fire rescue in the tunnel. The results show that:

(1) The developed FSDS is able to eliminate fire smoke particles and CO, cool the space, and improve the visibility of the fire site.

(2) The real fire experiment demonstrates that the FSDS operating performance parameters are as follows: $F_c = 8629.01 \text{ Nm}^3/\text{h}$, $F_n = 5472.37 \text{ Nm}^3/\text{h}$, $P_c = 417.7 \text{ KW}$, and $\eta_{CO} = 78\%$.

(3) The fire simulation in the rectangular tunnel shows that:

The FSDS exhaust forms the induced longitudinal airflow (20,000 m$^3$/h) along the length of the tunnel. In the FSDS operation process in the rectangular tunnel, the height of the smoke layer near the fire source is about 5.2 m, and the spray water will not destroy the structure of the smoke layer in the tunnel. In the FSDS operation process in the rectangular tunnel, the temperature in most of the tunnel area is lower than 60 °C. The CO concentration near the fire source is about 145 ppm. The visibility at the fire source is the lowest, and the value in other areas is greater than 50 m. The FSDS has a similar space cooling ability as the semitransverse ventilation, with the exhaust volume of 190,000 m$^3$/h. The FSDS has less damage to smoke layer structure than the semitransverse ventilation system.

The novel mobile fire smoke decontamination process and system still has several drawbacks, such as the lack of temperature control in the CO filtration process and small smoke decontamination capacity. Further promotion will be conducted in future study.

**Author Contributions:** Conceptualization, H.Y. and T.C.; methodology, Y.Z.; software, F.Z.; validation, F.Z.; formal analysis, F.Z.; investigation, L.H.; resources, Y.Z.; data curation, Y.Z.; writing—original draft preparation, F.Z.; writing—review and editing, F.Z.; visualization, F.Z.; supervision, L.H.; project administration, L.H.; funding acquisition, Y.Z. All authors have read and agreed to the published version of the manuscript.

**Funding:** This research received no external funding.

**Acknowledgments:** This work is supported by the study on freezing failure mechanism and performance of composite phase change PV/T solar collector system (2108085QE242); research and industrialization of fire smoke decontamination robot in long and narrow space based on information fusion technology (201903a05020024).

**Conflicts of Interest:** The authors declare no conflict of interest.

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
