# Peer review of "Development and Experimental Study of Mobile Fire Smoke Decontamination System"

_fire, doi:10.3390/fire6020055_

Round 1

Reviewer 1 Report

1. What is the smoke inlet scale? How to ensure that all smoke can be collected? As shown in Fig. 8, most of the smoke is not collected. Moreover, the physical size of the fire source in this paper is too small. However, the size of the fire source in the actual fire is relatively large, such as a car fire, which may result in the further diffusion of smoke. This paper needs to further improve the experiment for verification.

2. What is the heat release rate of the fire source in the experiment? It should be stated in the paper.

3. What is the heat radiation intensity that the robot can withstand? As shown in Fig. 8, the robot needs to be carried out near the fire source. In the long-narrow confined space, the heat radiation intensity near the fire source is relatively high due to the vertical flame and high-temperature smoke gas below the ceiling.

4. Grid size is an important index to affect the modelling accuracy. The grid sensitivity study must be carried out to show that the mesh selected in this paper is reasonable.  

5. Verification of FDS modelling should be carried out, that is, a comparison between the simulation results and the experimental results should be added to illustrate the accuracy of the simulation results in this paper.

6. The robot is placed near the fire source in the simulation scenario. How does the robot reach the fire source and discern the location of the fire source? In an actual fire scenario, there is a lot of smoke near the fire source.

7. Jet fans are generally used in the longitudinal ventilation system. The ventilation system studied in this paper is the semi-transverse smoke exhaust system. This must be revised.

8, In this paper, the fire size of 4 MW was selected in the simulation study. What is the reason? What is the smoke exhaust effect for a fire scenario with a larger fire size, such as 6 MW?

9. This paper states that the FSDS exhaust forms the induced longitudinal airflow (20000 m3/h). How to obtain the induced longitudinal airflow? Since air supply for combustion and smoke entrainment will also form the induced longitudinal airflow in the tunnel. The conclusion of this paper is not rigorous. It is necessary to further calculate the induced longitudinal airflow formed only by the fire (without ventilation) and then obtain the induced longitudinal airflow formed by the robot.

Author Response

Reviewer #1:

  1. What is the smoke inlet scale? How to ensure that all smoke can be collected? As shown in Fig. 8, most of the smoke is not collected. Moreover, the physical size of the fire source in this paper is too small. However, the size of the fire source in the actual fire is relatively large, such as a car fire, which may result in the further diffusion of smoke. This paper needs to further improve the experiment for verification.

Response: Thank you very much for the advice. For the first question, the diameter of the smoke inlet is 1000mm, the description has been added in fig. 4. The smoke inlet can collect about 70% of the fire smoke. Other 30% will be finally collected and decontaminated in subsequent smoke flow cycles. For the second question, in the case of car fire smoke decontamination (relatively high fire power), the FSDS mainly collect and decontaminate ceiling smoke in long and narrow spaces, which can effectively increase the height of the ceiling smoke. Further improvement of the experimental verification will be conducted in the future.

2: What is the heat release rate of the fire source in the experiment? It should be stated in the paper.

Response: Thank you very much for the advice. The oil pool fire and automobile tire combustion are adopted to simulate the fire occurrence process. The heat release rate of the fire source is in the range of 1~1.5 MW. This description has been added in the part of 3.1.

  1. What is the heat radiation intensity that the robot can withstand? As shown in Fig. 8, the robot needs to be carried out near the fire source. In the long-narrow confined space, the heat radiation intensity near the fire source is relatively high due to the vertical flame and high-temperature smoke gas below the ceiling.

Response: Thank you very much for the suggestion. The FSDS robot mentioned in this work is a prototype system. Heat shield and thermal insulation will be added in further promotion of the FSDS system.

  1. Grid size is an important index to affect the modelling accuracy. The grid sensitivity study must be carried out to show that the mesh selected in this paper is reasonable.

Response: Thank you very much for the suggestion. Non-uniform mesh gradually sparse from wall to interior area is used for mesh division in the theoretical model. The average grid size of the cross-section area of the tunnel is 20mm, and the average grid size of the longitudinal direction of the tunnel is 100mm. This description has been added in the part of 4.1.

  1. Verification of FDS modelling should be carried out, that is, a comparison between the simulation results and the experimental results should be added to illustrate the accuracy of the simulation results in this paper.

Response: Thank you very much for the advice. In this work, the FDS model is used for the prediction of the FSDS operation process and longitudinal ventilation process in the rectangular tunnel. In the model, the working performance parameters of FSDS is drawn from the test in this work. Further verification of this FDS modelling will be conducted in the future work (real tunnel fire experiment with FSDS).

  1. The robot is placed near the fire source in the simulation scenario. How does the robot reach the fire source and discern the location of the fire source? In an actual fire scenario, there is a lot of smoke near the fire source.

Response: Thank you very much for the advice. The robot reaches the fire source through remote control in an actual fire scenario.

  1. Jet fans are generally used in the longitudinal ventilation system. The ventilation system studied in this paper is the semi-transverse smoke exhaust system. This must be revised.

Response: Thank you very much for the advice. All “longitudinal ventilation system “in the paper have been replaced with “semi-transverse ventilation system”.

8, In this paper, the fire size of 4 MW was selected in the simulation study. What is the reason? What is the smoke exhaust effect for a fire scenario with a larger fire size, such as 6 MW?

Response: Thank you very much for the advice. As we mentioned in 4.1, the fire source (4 MW) is located in the center of the tunnel (Fire power of single car). The FSDS is design to deal with the fire smoke exhausted by a single car fire in underground garage or highway tunnel. In the case of 6 MW heat release rate, the flux of exhausted smoke will be larger.

  1. This paper states that the FSDS exhaust forms the induced longitudinal airflow (20000 m3/h). How to obtain the induced longitudinal airflow? Since air supply for combustion and smoke entrainment will also form the induced longitudinal airflow in the tunnel. The conclusion of this paper is not rigorous. It is necessary to further calculate the induced longitudinal airflow formed only by the fire (without ventilation) and then obtain the induced longitudinal airflow formed by the robot.

Response: Thank you very much for the advice. The induced longitudinal airflow is calculated through velocity integral of tunnel cross-section area. In this work, the induced longitudinal airflow is calculated to evaluate the working effect of the robot. Further mechanism study about the induced longitudinal airflow in tunnel will be conducted in future work.

Reviewer 2 Report

In this paper,  a novel mobile fire smoke decontamination process  and system are introduced. The experimental study and theoretical prediction of the sys tem are also conducted. The study is very useful for the fire safety engineering. For me, the paper can be accepted after minor revision. 

The authors need to check the figures. The resolution is too low. Besides, some minor errors exist in the description, which should be revised. 

Author Response

Reviewer #2:

In this paper,  a novel mobile fire smoke decontamination process  and system are introduced. The experimental study and theoretical prediction of the sys tem are also conducted. The study is very useful for the fire safety engineering. For me, the paper can be accepted after minor revision.

The authors need to check the figures. The resolution is too low. Besides, some minor errors exist in the description, which should be revised.

Response: Thank you very much for the advice. All figures and description in the paper have been checked and revised.

Reviewer 3 Report

1. The height of the smoke inlet should be described. Because the height of the smoke inlet will affect the efficiency of smoke ventilation.

2. After a fire occurs in the tunnel, the smoke first reaches the top of the tunnel and then spreads horizontally along the top of the tunnel to both sides. The optimal design requirements of smoke vent need to be further analyzed. For example, why the smoke vent opens down instead of up or sideways.

Author Response

Reviewer #3:

  1. The height of the smoke inlet should be described. Because the height of the smoke inlet will affect the efficiency of smoke ventilation.

Response: Thank you very much for the advice. As we noted in fig.4, The height of the smoke inlet is 1550mm.

  1. After a fire occurs in the tunnel, the smoke first reaches the top of the tunnel and then spreads horizontally along the top of the tunnel to both sides. The optimal design requirements of smoke vent need to be further analyzed. For example, why the smoke vent opens down instead of up or sideways.

Response: Thank you very much for the advice. The robot is designed to decontaminate the fire smoke in the early stage of the fire. In the early stage, the downward smoke inlet can create a downward umbrella-shaped negative pressure area, which can conduct an overlay inducement of the rising smoke flow.

Reviewer 4 Report

This paper proposed a novel fire smoke decontamination method and system. Relevant experimental and theoretical studies are conducted. The work is interesting and the results are valuable for further investigation on effective and developed fire smoke decontamination methods. I think this paper can be published in the present format after the following questions are answered.

1. Regarding The mixed cooling heat exchanger, how to ensure the portion (70%~80%) of the water spay evaporated into water vapor?

2.  What is the optimal operating temperature of the CO filter in FSDS?

3. The English presentation needs to be improved.

4. This study looks like a technical note, lacking theoretical analysis.

Author Response

Reviewer #4:

This paper proposed a novel fire smoke decontamination method and system. Relevant experimental and theoretical studies are conducted. The work is interesting and the results are valuable for further investigation on effective and developed fire smoke decontamination methods. I think this paper can be published in the present format after the following questions are answered.

  1. Regarding The mixed cooling heat exchanger, how to ensure the portion (70%~80%) of the water spay evaporated into water vapor?

Response: Thank you very much for the advice. The structure and water flux of the mixed cooling heat exchanger of the robot are optimal designed to deal with the 200℃ inlet fire smoke, which can ensure the portion (70%~80%) of the water spay evaporated into water vapor.

  1. What is the optimal operating temperature of the CO filter in FSDS?

Response: Thank you very much for the advice. The optimal operating temperature of the CO filter in FSDS is 150~200℃。

  1. The English presentation needs to be improved.

Response: Thank you very much for the advice. The English presentation of the paper has been improved again.

  1. This study looks like a technical note, lacking theoretical analysis.

Response: Thank you very much for the advice. The paper is submitted as a technical note.

Round 2

Reviewer 1 Report

1.The grid sensitivity study should be carried out to show that the mesh selected in this paper is reasonable. 

Author Response

  1. The grid sensitivity study should be carried out to show that the mesh selected in this paper is reasonable.

Response: Thank you very much for the advice. The grid sensitivity test has been conducted through choosing a typical simulation condition. The results are show in table 4. It can be seen that the selected mesh in the model is able to draw the simulation results with high accuracy.

Reviewer 4 Report

Agree to accept in this version.

Author Response

Thank you very much for the advice. The English language and expression have been revised again.